# Endogenous and Exogenous Regulation of Redox Homeostasis in Retinal Pigment Epithelium Cells: An Updated Antioxidant Perspective

**DOI:** 10.3390/ijms241310776

**Published:** 2023-06-28

**Authors:** Yuliya Markitantova, Vladimir Simirskii

**Affiliations:** Koltsov Institute of Developmental Biology, Russian Academy of Sciences, 119334 Moscow, Russia

**Keywords:** retinal pigment epithelium, redox homeostasis, reactive oxygen species, oxidative stress, endogenous cell defense, redox-sensitive transcription factors, target genes, antioxidant therapy

## Abstract

The retinal pigment epithelium (RPE) performs a range of necessary functions within the neural layers of the retina and helps ensure vision. The regulation of pro-oxidative and antioxidant processes is the basis for maintaining RPE homeostasis and preventing retinal degenerative processes. Long-term stable changes in the redox balance under the influence of endogenous or exogenous factors can lead to oxidative stress (OS) and the development of a number of retinal pathologies associated with RPE dysfunction, and can eventually lead to vision loss. Reparative autophagy, ubiquitin–proteasome utilization, the repair of damaged proteins, and the maintenance of their conformational structure are important interrelated mechanisms of the endogenous defense system that protects against oxidative damage. Antioxidant protection of RPE cells is realized as a result of the activity of specific transcription factors, a large group of enzymes, chaperone proteins, etc., which form many signaling pathways in the RPE and the retina. Here, we discuss the role of the key components of the antioxidant defense system (ADS) in the cellular response of the RPE against OS. Understanding the role and interactions of OS mediators and the components of the ADS contributes to the formation of ideas about the subtle mechanisms in the regulation of RPE cellular functions and prospects for experimental approaches to restore RPE functions.

## 1. Introduction

Restoring the functions of retinal pigment epithelium (RPE) cells and retinal neurons, which are characterized by high metabolic activity, is an urgent problem in biomedicine and ophthalmology. The RPE is critical for the homeostasis and function of retinal neurons [1] (Figure 1). The identification of the mechanisms of RPE cell homeostasis disturbance in neurodegenerative pathologies and determining ways to prevent such pathologies are priorities of modern research. Studies of the biology of vertebrate RPE cells over the past two decades have made significant adjustments to the fundamental knowledge of the molecular and epigenetic mechanisms in the regulation of developmental and functional processes [2,3,4,5]. RPE functions are regulated as a result of the coordinated interaction between different levels of the endogenous cell defense system [6,7,8]. The RPE has a regulatory network for maintaining homeostasis, which ensures the integrity of the proteome and enables adaptation to changes in response to the action of endogenous and exogenous factors. Reparative systems of endogenous cell defense include main mechanisms such as the remodeling of damaged proteins, their restoration and/or degradation by the ubiquitin–proteasome system or autophagy, and antioxidant protection [9,10,11]. Maintaining a balance between oxidative and reduction processes (redox balance) ensures RPE homeostasis. Reactive oxygen species (ROS) and antioxidants are the main functionally related components of redox processes [12]. Under normal conditions, a balance between oxidative and reductive processes (redox homeostasis) is maintained in cells. Disbalance in the direction of oxidative processes leads to oxidative stress (OS) [13,14]. OS is a leading factor in the pathogenesis of the RPE and associated degenerative retinal diseases, including age-related macular degeneration (AMD), proliferative vitreoretinopathy (PVR), diabetic retinopathy, retinitis pigmentosa, and uveitis [15]. The processes of free-radical oxidation are enhanced when the integrity and homeostasis of tissues are violated, causing the development of OS, which, together with inflammatory factors, disrupts the work of a number of regulatory factors and enzyme systems [16]. Structural metabolic and genetic changes in the RPE and border tissues are accompanied by the accumulation of toxic components, which leads to a shift in redox homeostasis towards OS development [4,17,18]. The RPE response to OS may mediate retinal damage or survival, depending on the severity and duration of OS. The introduction of genomics and proteomics methods enriches our understanding of the molecular participants that maintain the stability of RPE differentiation, the mechanisms of autophagy cell protection, protein proteolysis, and apoptosis regulation, which underlies the development of methods for targeted therapy and prevention in regard to the development of OS-dependent degenerative retinal diseases. Studies of the transcriptome, proteome, and metabolome of RPE cells in in vivo and in vitro models have been fruitful in identifying a number of redox-dependent genes (transcription factors) and proteins involved in the control of the differentiation of RPE cells and the choroid and the regulation of homeostasis [19,20,21,22].

The role of a large number of factors and regulators of RPE functions, characterized by active metabolism, has been revealed [23,24]. Most of the genes in which mutations lead to retinal diseases are genes with a high and specific expression in the RPE or photoreceptors. Genetic disorders that affect the functioning of transcription factors involved in the control of differentiation in embryogenesis lead to RPE dysfunction, the death of neurons in the outer nuclear layer and in photoreceptors (rods and cones), and the death of neurons in the inner nuclear layers of the retina [25,26,27]. Transcriptome analysis showed a high level of expression of regulatory redox-dependent genes that provide resistance to oxidative reactions in the RPE [28,29,30,31]. RPE homeostasis largely depends on the stability state of the Bruch’s membrane (BM) [1]. BM and photoreceptors are susceptible to high levels of OS, especially in the macula region of the retina, where incoming light is focused and a high density of cone photoreceptors is observed [32]. A recent study of the RPE transcriptome in the macula and periphery of the retina, obtained using microarray technology, revealed differential expression of extracellular matrix (ECM) protein genes in MBs, which reflects the functional state of this structure [33]. ECM remodeling and changes in the BM composition can disrupt the diffusion of oxygen and macromolecules and thus cause hypoxia [23], which is an important link in triggering RPE dysfunction and neurodegenerative processes in retinal photoreceptors [34,35].

We performed a comprehensive analysis of the current literature on endogenous defense systems in the RPE with a focus on antioxidant systems, identifying current gaps and problems in antioxidant therapy within this field of study. This review systematizes information about the key links in the regulation of redox metabolism and the antioxidant protection of the RPE, as well as information about their relationship with important links in the overall endogenous protection of cells. Data obtained through in vivo and in vitro models using modern methods of transcriptomics and genome editing are analyzed. The prospects and limitations of the developed therapeutic approaches that aim to restore redox balance and general intracellular homeostasis in RPE cells are considered, with a focus on those that impact the key molecular links of the ADS.

## 2. Functions of the RPE

The RPE is a single-row hexagonal layer of polarized pigmented cells. RPE cells on the apical side form tight contacts with the photoreceptors’ outer segments (POSs), and on the basal side they interact with the MB, which is closely associated with the choroid [2,20]. The physiological functions of RPE cells are associated with protecting (shielding) photoreceptors from excess light (Figure 1). Additionally, the RPE maintains homeostasis, pH, and circulating liquid volume in the subretinal space by transporting metabolites to the choroid [1,36]. As a result of the reactions of the visual cycle and the metabolism of rhodopsin, during the isomerization from 11-cis-retinal to trans-retinal, a large amount of ROS is produced, causing lipid peroxidation (LPO) [37,38]. In the retina, there is a constant renewal of photoreceptor disks. RPE cells phagocytize used POSs, which subsequently leads to autophagy-lysosomal degradation. The RPE renews “exhausted” photoreceptor disks enriched in ROS and lipofuscin (LF), the main product of LPO [39]. Changes in the light regime affect intracellular pH and the concentration of Ca^2+^ and K^+^ ions and increase oxygen consumption, causing the generation of CO_2_ and H_2_O in the RPE [40]. The RPE is involved in maintaining retinal homeostasis by transporting glucose to neurons using GLUT-1 and -3 transporters [1,41], maintaining the pH and volume of the circulating fluid in the subretinal space and transporting metabolic products to the choroid [42,43,44]. The RPE and choroid secrete a number of growth factors including VEDF, PEDF, TGFβ [45], interleukins IL-6 and IL-10, tumor necrosis factor-beta (TNF-β), interferon-β, complement factor H, monocytic chemotactic protein-1 (MCP1), and other factors that are involved in the regulation of RPE homeostasis and the immune response in pathologies [21,46,47,48]. TNFα is involved in maintaining the secretion and level of VEGF and, in cooperation with TGFβ2 and FGFβ, the induction of VEGF expression, thus maintaining the constancy of RPE cell homeostasis [49]. The function of PEDF is to suppress the development of a local inflammatory process as a result of the modulation of macrophage activity [50]. The balance of growth factors is important for ensuring the functions of the RPE and photoreceptors [4,51].

BM underlies the RPE from the choroid side and plays an important role in regulating the transport of biomolecules (proteoglycans, chemokines, cytokines, growth factors, and toxic waste products) between photoreceptors, the RPE, and the choroid [52]. BM is a dense, cell-free fibrillar layer of the proper RPE basal lamina that is rich in collagen and elastin, with a predominance of heparan sulfates. BM also includes endothelial components of the capillary-rich choroid [53]. The formation of excessive amounts of basement membrane components may be a general epithelial cell response to stress in order to remain adherent. The disruption of ECM homeostasis likely results in an environment of increased OS that contributes to disease onset and progression [54]. BM provides a mechanical function, enables cell adhesion, acts as a barrier that limits the migration of choroid cells, and ensures the migration and differentiation of RPE cells in embryogenesis. In MB, these processes are strictly regulated for maintaining local RPE and retinal homeostasis, which depends both on the genetically determined state of tissues and on the influence of exogenous factors [23,55,56].

The photoreceptor–RPE–BM–choroid complex underlies the formation of the blood–retinal barrier. The RPE together with BM is involved in the formation of the outer blood–retinal barrier [57]. As a component of the outer part of the blood–retinal barrier, the RPE is involved in the creation of the immune privilege of the retina, mainly due to tight intercellular contacts [58,59].

RPE cell dysfunction contributes to the development of retinal diseases, including AMD, PVR, and glaucoma [26,60,61]. Despite the differences in the etiopathogenesis of these diseases, one of the common trigger mechanisms for their development is a violation of redox homeostasis in RPE cells.

## 3. Functional Prerequisite of RPE Cells for Oxidative Stress

The RPE is characterized by its tendency to have high oxygen tension due to its close proximity to the choriocapillaris and is considered to be exposed to the most intensive O_2_ pressure of any human tissue. The high metabolism rate of the RPE and retinal photoreceptors, the constant renewal of the membrane disks of POSs by phagocytosis, the high levels of polyunsaturated fatty acids and exposure to light, and the high oxygen consumption and intensity of energy metabolism processes are necessary to maintain normal physiological function [62]. ROS occur in photoreceptors as a result of the activity of mitochondria (as a byproduct of the oxidative phosphorylation chain) and due to the process of phototransduction under the action of light on the light-sensitive pigments of rods and cones. In turn, the peroxidation of polyunsaturated fatty acids (docosahexaenoic acid, etc.) that enrich the plasma membranes of photoreceptor discs occurs under the action of ROS. The RPE has a large number of mitochondria due to its high energy requirements and elevated metabolic rate, which is a prerequisite for the production of high levels of ROS, which are the products of oxidative phosphorylation [3].

Intense mitochondrial metabolism, the phagocytosis of POSs, the phototoxic activity of LF, and the photosensitization of hemoglobin precursor are the sources of free radicals not only in the RPE but also in photoreceptors [63,64] (Figure 2).

In postmitotic cells, the redox balance in the RPE largely depends on autophagic clearance and the intensity of accumulation of cellular debris, including LF. With age, this balance shifts towards the intensification of oxidative reactions, which enhances the production of ROS [65]. The accumulation and deposition of LF in lysosomes occur throughout one’s life, which is a sign of the natural aging of the RPE. Lysosomes become incapable of completely breaking down all oxidized products of photoreceptor metabolism. This leads to the ferritin-mediated formation and accumulation of iron and non-degradable LF pigment in RPE lysosomes [39,66]. At the same time, LF can occupy up to 25% of the volume of RPE cells, causing residual molecular components to accumulate in the cytoplasm and impairing their export through the plasma membrane [67]. The dysfunction of RPE cells and impaired photoreceptor metabolism is due to excessive accumulation of LF. The release of lysosomal-degrading enzymes into the cytosol causes cell death [65,68]. Under OS conditions, the phagocytic load of RPE cells increases, which manifests itself in the increased phagocytosis of apoptotic fragments of dying cells and the utilization of BM metabolic products [39].

Genetically determined defects causing the intensification of LF accumulation in the RPE, a decrease in concentration and/or the complete absence of melanosomes, and pathological BM remodeling are important links in RPE dysfunction [69,70]. A decrease in the number of melanosomes in human RPE—which perform screening and antioxidant functions—as a result of the destruction of melanin by superoxide radicals generated by LF under the action of visible light is associated with cellular aging. These processes serve as a prerequisite for the intensification of oxidative reactions and OS [71]. A significant increase in the concentration of LF granules, which correlates with a decrease in the concentration of melanin-containing pigment granules, is observed in RPE cells in AMD [72]. An inverse correlation was found between the process of LF formation and the inhibition of proteasome activity in the RPE [73]. A decrease in proteasome activity leads to the accumulation of LF in the RPE and is associated with a decrease in the intensity of autophagy [10,74,75]. RPE dysfunction and damage and, as a result, retinal degeneration, can be caused by systemic diseases [76]. Many works have been devoted to studying the mechanisms of oxidative damage in neurodegenerative pathologies of the retina in Alzheimer’s and Parkinson’s disease [77,78]. OS leads to the accumulation of glycation end products and lipid and sugar oxidation, which, in turn, contributes to the production of ROS and reactive forms of nitrogen and enhances damage to RPE cells. OS-dependent chronic hyperglycemia is an important factor in the pathogenesis of diabetic retinopathy [65]. The destructive activity of ROS manifests itself in the oxidation of proteins and membrane lipids and can lead to DNA damage [8]. With excessive accumulation, oxidized forms of proteins lose their inherent structure and aggregate, which can further promote endoplasmic reticulum (ER) stress and cause a reaction in unstructured proteins. The accumulation of aggregated proteins causes even more OS, leading to lysosome rupture and inflammation [79,80]. Free-radical oxidation is reinforced as tissue integrity and homeostasis are affected, and it is then transformed into OS, which cooperates with inflammatory factors to impair the function of certain regulatory factors and enzyme systems in the RPE [16,81].

The excessive accumulation of advanced glycation end products enhances the sensitivity of the RPE to pro-inflammatory stimuli, activates the transcription factor NF-κB, and leads to pathological ECM remodeling [82]. Hyperglycemia is accompanied by an intracellular decrease in the concentration of H^+^ ions, which results in acidification of the retina. Growth factor imbalance, increased production of pro-inflammatory cytokines and ROS [42,83], ATP secretion, and activation of ATP-dependent Ca^2+^ channels (P2X7R) are signals of damage to RPE and retinal cells [31,84].

Interactions between the RPE and neighboring tissues maintain local RPE and retinal homeostasis, which depends on both the genetically determined states of the interacting tissues and exogenous factors [23]. Structural, metabolic, and genetic disorders of the RPE and neighboring tissues lead to the accumulation of toxic components, which poses a risk for redox disbalance, leading to OS [62,63,64]. These endogenous factors predispose RPE cells and photoreceptors to OS. The sensitivity of RPE cells and photoreceptors to OS, together with their exposure to exogenous factors including excessive light, ionizing radiation, high temperature, certain chemicals (e.g., nitrates), and mechanical injury [85,86] increase the risk of oxidative damage. Under blue light exposure, intense OS occurs in the POS, the region which the RPE “cleanses” (including the removal of the end products of the visual cycle), and this forms an important source of accumulated waste.

## 4. The Maintenance of Redox Homeostasis in RPE Cells

### 4.1. Key Components of the Pro-Oxidant System in the RPE

Here, the main elements of the pro-oxidant system in RPE cells are briefly described (Figure 2). The pro-oxidant system of RPE cells includes ROS [87]: hydroxyl radical (OH∙), singlet oxygen (^1^O_2_), superoxide (O_2_^−^), hydrogen peroxide (H_2_O_2_), and specialized enzymes, including NADP oxidase, nitric oxide synthase, and xanthine oxidase [88,89,90,91,92].

#### ROS Sources in RPE

The main sources of ROS in the body are phagocytes: granulocytes, monocytes, macrophages, neutrophils, and eosinophils [93]. ROS (superoxide radicals) appear in the RPE during ER stress as a byproduct of oxidative phosphorylation, which occurs largely in the mitochondria. Superoxide radicals are massively generated in dark processes of the mitochondrial respiratory chain, as well as by exposure to visible light [94,95].

The production of ROS in mitochondria is highest at high mitochondrial membrane potentials. The increased ROS level activates uncoupling proteins located in the mitochondrial inner membrane and increases the transport of protons from the mitochondria, thereby reducing its membrane potential and the production of ROS [96]. Uncoupling proteins (UCPs) are a part of the large family of mitochondrial solute carriers. In addition to UCP2, the 2-oxoglutarate and dicarboxylate carriers were recently identified in RPE mitochondria [97]. These carriers transport glutathione into mitochondria and exhibit a time- and dose-dependent decrease with stress [98].

It is difficult to estimate the relative contributions of ROS due to the environment versus those produced due to the high metabolic flux through the electron transport chain. ROS production is dependent on proton leaks that consist of a basal proton leak and induced proton leak. The basal proton leak is unregulated and correlates with the metabolic rate. The induced proton leak is precisely regulated in stress conditions and is induced by superoxide or peroxidation products through UCPs [99,100]. Therefore, the contribution of exogenous stressors to the production of mitochondrial ROS can be indirectly judged by the activity of UCPs.

Mitochondrial OS in the RPE leads to metabolic dysfunction in both the RPE and retinal photoreceptors [95,101]. Mitochondrial DNA is more susceptible to oxidative damage in neuropathologies [102]. The excessive production of ROS in mitochondria correlates with age-related DNA disorders [103]. The action of ROS causes LPO of, for example, docosahexaenoic acid, among others, which can be enriched in the plasma membranes of photoreceptor disks [94,104]. LF granules also serve as a source of photo-induced generation of superoxide radicals in RPE cells; LF exposed to visible light (largely in the blue region) reduces oxygen to superoxide radicals [105]. The cytoplasmic metalloprotein xanthine oxidase is involved in the catabolism of purines, catalyzing the conversion of hypoxanthine to uric acid with the production of superoxide anions [13].

Other endogenous sources of ROS formation are the enzyme system of the transmembrane complex of NADPH oxidases (NOXs), monoamine oxidases, and NO synthases, which generate superoxide anions and hydrogen peroxide and are involved in the OS reaction aimed at destroying pathogens [106,107,108]. Various NOX isoforms are found on the plasma membrane, nuclear membrane, and membranes of the ER, mitochondria, and phagosomes, where they are produced and participate in cellular regulatory (kinase) cascades, regulate ion transport systems, and can activate transcription factors [109,110,111,112,113].

ROS (hydroxyl anion) can be generated in lysosomes as a result of the reaction between superoxide and nitric oxide [114].

Another source of ROS is peroxisomes where fatty acids are β-oxidized to form H_2_O_2_ and ^1^O_2_ [62]. Singlet oxygen (^1^O_2_) generated by photosensitization and the excitation of LF, riboflavin, or the retina is also a deleterious oxygen metabolite for the cell. At the same time, both ^1^O_2_ and HO• are aggressive but unstable and rapidly react with membranes and other cellular compartments [115].

### 4.2. Key Components of the ADS in RPE Cells

The regulatory hierarchy of the main levels of the ADS is universal and responsible for the effective neutralization of free ROS, as well as for the reduction in oxidized molecules. The endogenous system of antioxidant defense in RPE cells also includes several regulatory levels (Figure 3). The first level is largely provided by the coordinated activity of redox- and OS-sensitive transcription factors.

#### 4.2.1. Transcription Factors

The ADS of RPE cells includes transcription factors that play an important role in the mechanisms of maintaining cell redox homeostasis to reduce intracellular OS and prevent age-related cell pathologies [116]. Redox- and OS-responsive transcription factors that control key processes (autophagy, cell proliferation, death, and other reparative processes) are critical components of the signaling pathways for the regulation of redox homeostasis and activation of endogenous RPE cell protection. Redox-sensitive factors that are the first line of cell protection in the RPE are the specific ROS sensors that trigger the ADS.

Redox-sensitive protective factors that are the first line of cell protection in the RPE include the transcription factors Nrf2, FoxO, HSF-1, AP-1, etc., which control the expression of antioxidant genes [116,117,118], in addition to nuclear transcription factor NF-kB [119]. A key factor in the antioxidant cellular response is the nuclear factor erythroid 2-related factor 2 (Nrf2). The role of Nrf2 in the activation of autophagy in the RPE has been demonstrated in knockout mice that model some of the properties (drusen formation) characteristic of the AMD [120]. Nrf2 regulates the protein turnover by inducing proteasome subunits to confer protective effects against chronic diseases and modifies cellular metabolic processes such as, for example, the pentose phosphate pathway, which provides NADPH and purine nucleotides that are essential for redox homeostasis and cellular proliferation. Nrf2 controls genes associated with energy metabolism, DNA repair, and autophagy regulation. The targets of action in the RPE for the transcription factor Nrf-2 cover a wide range, including components of both antioxidant and anti-inflammatory signaling cascades [121,122,123].

Nrf2 binds to antioxidant response sites (AREs) in the promoter regions of target genes. This regulatory sequence is usually found upstream of genes encoding phase II detoxifying enzymes and has been known to regulate the induction of these genes [124]. The main function of phase II enzymes is to detoxify the highly reactive intermediate metabolites generated by phase I reactions and accelerate the excretion of toxic xenobiotics. The downregulation of the expression of phase II enzymes in Nrf2-knockout mice indicated that Nrf2 regulates the global transcription of phase II enzymes through ARE-dependent signals [109,125].

The discovery of AREs has led to the conclusion that the battery of genes encoding antioxidant and cytoprotective proteins, including glutamate-cysteine ligase (GCL), thioredoxin reductase 1 (Txnrd1), NAD(P)H-quinone oxidoreductase 1 (NQO1), and heme oxygenase-1 (HO-1), is regulated through Nrf2 binding to this consensus-binding sequence [124,125].

Beyond the *Nrf* gene family, there are other redox-sensitive transcription factors such as AP-1, NFκB, p53, and HIF that may be involved in generating chemical-specific effects. PPAR, CREB, TFEB, Foxo 1/3, and FXR [117], as well as autophagy-associated proteins ATG7 and Beclin-1, are also among the redox-sensitive transcription factors involved in the regulation of metabolism and autophagy [9,126,127,128]. BECN1, ATG7, and ATG9 are constitutively expressed in RPE cells; these proteins are critical for autophagy, and their activation depends on OS intensity and duration. Acute OS intensifies autophagy; however, chronic OS depletes these proteins in ARPE-19 cells and reduces autophagy [9]. OS impact is mediated by transcription factors-1 (ATF-1) of the ATF family and Fos family transcription factors that control cell proliferation, autophagy, and apoptosis [125].

In RPE cells, Nrf2 acts in cooperation with the transcription factor MITF (microphthalmia-associated transcription factor), and this relationship is reciprocal [129]. MITF has been demonstrated to regulate RPE development and differentiation, melanogenesis, migration, proliferation, growth factor secretion, and visual cycle function [130]. Interactions between these transcription factors control melanogenesis, mitochondrial biogenesis, and redox homeostasis [131].

MITF directly binds to the *NRF2* promoter and promotes its nuclear translocation, predominantly through upregulating p62 [129]. MITF also regulates other target genes in different functional pathways; for example, the combination of NRF2 with other MITF downstream factors, such as PEDF and PGC1α, might lead to a more complete redox signaling in RPE cells and can regulate mitochondrial biogenesis [131,132].

H_2_O_2_ reduces the expression level of MITF but increases that of Nrf2 in the nucleus. As a consequence, the expression levels of antioxidant enzymes in H_2_O_2_-treated cells are upregulated, but the expression levels of proteins involved in melanin synthesis are downregulated [133].

Exposure to light and oxygen and oxidative phosphorylation processes generate ROS through the mitochondrial electron transport chain. ROS can be restored with the help of NADPH, glutathione, and antioxidant enzyme systems [109,134].

#### 4.2.2. Melanin

The pigment melanin performs the function of “neutralization” of ROS (singlet oxygen) [135,136], causes the oxidation of superoxide radicals into molecular oxygen, reduces superoxide to hydrogen peroxide [105,137], binds redox-active metal ions, and performs the function of photoprotection [138].

#### 4.2.3. Enzyme Systems That Neutralize and Reduce ROS

Enzyme systems that protect RPE cells from OS include OS-neutralizing and OS-reducing enzymes. OS induces the expression of phase II enzymes including NADPH quinine oxidoreductase 1, HO-1, and catalytic (GCLC) and modulatory (GCLM) subunits of glutamate-cysteine ligase [139,140].

The RPE and retina of a rat demonstrated activation of the antioxidant protection enzymes Cu/Zn superoxide dismutase (SOD), Mg SOD, catalase, glutathione peroxidase (GPX), and the catalytic subunit of glutamate-cysteine ligase that is linked to the regulation of endogenous cyclophilin B levels [141].

SOD is a metalloenzyme whose synthesis increases under OS. This enzyme catalyzes O2- and transforms it into H_2_O_2_. SOD is expressed in mitochondria and protects cellular and mitochondrial structures from superoxide [142,143]. It has been shown in adult mice that mutations in the *SOD1*^−/−^ and *SOD2*^−/−^ genes lead to the development of signs of neovascularization and the death of retinal neurons, resembling human AMD [144,145].

Exposure to light and oxygen and oxidative phosphorylation generate OS via the electron transport chain. These ROS can be reduced by employing the systems of NADPH, glutathione, and antioxidant enzymes. NADPH is the key reductive equivalent generated via the pentose phosphate pathway in glucose oxidation by serine synthesis from 3-phosphoglycerate and by the activities of the NADP^+^-malic enzyme or NADP^+^-isocitrate dehydrogenase [109,134]. NADPH and reduced glutathione (GSH) are the main reducing agents that create a buffer system that prevents OS development. NADPH is a cofactor in GSH reduction, peroxiredoxin metabolism, and the reduction of cystine to cysteine. GSH prevents protein oxidation and is also used as a substrate for GPX [146]. In this regard, it should be noted that NOX enzymes are involved in a variety of regulatory signaling cascades. The oxidation of redox-active cysteines at the active site of protein phosphatases is the best-studied mechanism of NOX action. ROS suppress their activity by increasing tyrosine phosphorylation in proteins. For instance, the protein tyrosine phosphatase 26 regulates the phosphorylation of protein components of cell proliferation, death, differentiation, and metabolic signaling pathways [147]. Superoxide is largely generated in mitochondria as a byproduct of cellular respiration as well as in the cytosol through NOX activity [148,149].

Catalase converts H_2_O_2_ that is generated after the β-oxidation of fatty acids in peroxisomes into water and oxygen [146]. The catalase content of RPE cells is several times that of other eye tissues [17].

GPX catalyzes the breakdown of H_2_O_2_ into water in the cytosol using GSH as a reducing agent [150]. A decrease in GPX activity in RPE cells and photoreceptors increases their sensitivity to OS [97,151]. GPX, structurally a selenium-containing glycoprotein, is a key enzyme that protects membrane cells under low OS from oxidative damage [146].

#### 4.2.4. Iron Chelators

Endogenous proteins with iron-binding properties in the RPE include ferritin, metallothionein, and some heat shock proteins (HSPs) [152,153]. Under physiological conditions, the levels of iron-binding proteins and HSP70 proteins in RPE cells are usually low, but they increase in response to OS and changes in intracellular pH, participating in the chelation of iron cations and the regulation of proteostasis [153,154]. The violation of iron metabolism and its accumulation in cells, which has a toxic effect, is associated with a decrease in autophagy and the work of chelating agents. Disturbed iron metabolism and toxicity due to accumulation in the cell is an important destabilizing factor of RPE differentiation and degeneration in retinal neurons. These processes relate to disturbed autophagy and the activity of chelating agents in the RPE. Concerning retinal diseases, the disruption of these processes occurs during the development of AMD [155,156]. Accumulated iron has been found in the RPE, photoreceptors, and BM in AMD patients [157].

#### 4.2.5. Chaperone Proteins

RPE cells constitutively contain the small HSP αB-crystallin, which can function as an antiapoptotic protein induced during OS [158]. The role of HSPs is to prevent the intracellular accumulation of cytotoxic proteins, regulate protein folding, and recover damaged proteins in lysosomes and peroxisomes [159,160]. Under OS conditions, when the level of ATP in RPE cells decreases, the ATP-independent chaperone HspB1 is among the first to be activated. In this case, the external receptor-dependent pathway of cell death is blocked with the participation of tumor necrosis factor receptors (TNFRs) and the internal mitochondrial signaling pathway. After partial cell recovery, the ATP-dependent chaperone HSP70 is activated [161]. Hsp70 inhibits the formation of a functional apoptosome by direct interaction with Apaf-1, protects against the forced expression of caspase 3, and prevents the translocation of Bax from the cytoplasm to the mitochondria [154,162]. Hsp90 can prevent the formation of the apoptosome complex by inhibiting the oligomerization of Apaf-1 [154]. It maintains the activity of Akt by inhibiting its dephosphorylation. Akt is activated by VEGF and NPD1, and both are induced by OS in the RPE [49,163]. PI3K-Akt-mTOR pathway activation mainly occurs in the Nrf2-related response to OS [75,124,164].

Hsp27 can maintain mitochondrial stability and redox homeostasis in cells and interacts with the apoptotic signaling pathways at many stages. It can inhibit apoptosis through the sequestration of Bax and Bcl-xS in the cytoplasm and is also involved in the stabilization of Akt [159,160].

Functions of HspB1 in the RPE consist of the blocking signaling pathways that trigger caspase-dependent apoptosis [161]. The activation of the low-molecular-weight chaperone Hsp27 leads to the blocking of Ca^2+^-induced apoptosis, which is a result of the suppression of caspase-3 functions and the prevention of cytochrome C release from mitochondria into the cytoplasm [159,160]. OS induces the expression of redox-dependent antioxidants and DJ-1 chaperones in the RPE [165], such as alpha-1 microglobulin, that binds to ROS [166]. Under OS conditions, the antioxidant activity of αA- and αB-crystallins, belonging to the family of chaperone proteins, increases. αA-crystallins are involved in the activation of phosphorylation reactions in the PI3K/Akt signaling pathway, which ensures the resistance of RPE cells to OS [167]. In knockout mice with αA- and αB-crystallin genes under conditions of simulated OS, the accumulation of ROS by RPE cells was observed, which was followed by the degeneration of retinal photoreceptors [168].

#### 4.2.6. Low-Molecular-Weight Antioxidants

Water-soluble antioxidants such as ascorbic acid (vitamin C) and GSH act as ROS scavengers in the cytosol [169]. Fat-soluble antioxidants that include α-tocopherol (vitamin E) and carotenoids (β-carotene, lutein, zeaxanthin, and lycopene) are associated with lysosome and mitochondrial membranes and protect cells from LPO. α-Tocopherol is one of the strongest antioxidants due to its capacity to arrest the autocatalytic chain reaction of LP. Carotenoids rely on a similar mechanism to neutralize peroxyl radicals and ^1^O_2_ is able to interrupt the autocatalytic chain reaction of LPO [170,171].

#### 4.2.7. Additional Endogenous Neuroprotectors

The RPE synthesizes biomolecules such as neuprotectin D1, which can play a protective role [172]. The synthesis of neuprotectin D1 by RPE cells during phagocytosis ensures the resistance of these cells to OS [173].

Purinergic signaling cascades also contribute to the regulation of redox homeostasis in the RPE. The balance between extracellular ATP and adenosine maintains the pH of the lysosomes, and its disturbance changes the lysosomal activity of RPE cells and stimulates excess LF production [174,175].

The ADS of RPE cells regulates the operation of a spectrum of genes and proteins that maintain cell viability [119,176]. A balance between pro- and antioxidant systems in the RPE is maintained by the activity of signaling proteins and enzymes, as well as by redox-sensitive genes controlling DNA synthesis, the permeability of membrane channels, and other processes underlying cellular homeostasis [6,7,8].

Reparative autophagy, the remodeling of damaged proteins, and the ADS are critical processes in the RPE response to withstand OS. These evolutionarily conserved mechanisms are necessary for transcriptional responses and adaptations to stressful conditions [116]. The coordinated work of listing the regulatory links for the antioxidant defense of cells ensures the restoration of balance between the oxidative and reduction processes in the RPE.

#### 4.2.8. Autophagy

Cellular and molecular processes that control the production and elimination of ROS ensure the implementation of the main functions of RPE cells and their survival [12,177]. This section of the review considers the main interrelated components of the endogenous regulation of the redox balance in RPE cells. Redox homeostasis is a critical persistence factor for RPE cells to maintain the balance between their pro- and antioxidant systems. The regulation and maintenance of redox homeostasis make it possible to prevent or restore ROS-mediated damage to RPE cells.

ROS, along with other cellular stress factors (inflammation and exposure to toxins), cause an increase in reparative autophagy in the RPE, which protects RPE cells from oxidative damage and is aimed at restoring altered cellular structures [11]. In most cases in the RPE, cellular components damaged under the action of OS are restored or degraded and replaced by newly synthesized ones via autophagy, with the participation of lysosomes. Autophagy induction is the most important mechanism that ensures the resistance of RPE cells to adverse conditions, oxidative reactions, and toxic effects [1,178].

Metabolically active RPE cells normally demonstrate a high basal rate of autophagy compared to retinal neurons. However, this process becomes less efficient with age, which was found in human and mouse RPE [135,179]. An inverse relationship was found between a decrease in the rate of autophagy and the formation of drusen in RPE pathologies [180]. Many metabolic transcription regulatory factors are involved in the regulation of RPE autophagy, for example, PPAR, CREB, TFEB, Foxo 1/3, FXR [117], ATG7 protein, and Beclin-1 [126,127]. Proteins BECN1, ATG7, and ATG9 are constitutively expressed in the RPE; their activity is critical for the functioning of signaling pathways that trigger autophagy [9]. The blockade of autophagy by 3-methyladenine or the knockdown of the *ATG7* and *Beclin-1* genes enhances the oxidative and cytotoxic effects of H_2_O_2_ [9,126]. Conversely, the activation of autophagy protects RPE cells from OS induced by NaIO_3_ [181,182] or H_2_O_2_ [18].

The inhibition of autophagy induces cellular dedifferentiation mammal RPE—a phenomenon that is characterized by a reduction in the levels of RPE-specific proteins and by cellular hypertrophy, the induction of the epithelia–mesenchymal transition (EMT), fibrosis, and the degeneration of the RPE due to OS [26,60,61].

The main functionally linked redox-active molecular components are OS and antioxidants. The interaction between the pro-oxidant and antioxidant systems ensures the regulation of the equilibrium concentration of ROS and antioxidants [7,183]. It is known that various ROS are able to regulate the activity of intracellular enzyme systems (proteinases, phospholipases, and phosphatases), modulate the expression of genes encoding transcription factors, and influence cell metabolism [117,118,184].

## 5. Key Components in the Mechanisms of Oxidative Stress Realization in the RPE—Potential Molecular Targets

### 5.1. Intracellular and Molecular Targets of OS

OS is an important part of overall cellular stress, which is influenced by various endogenous factors. The destructive activity of ROS in cells is manifested in the damage and oxidation of molecular targets, such as DNA, proteins, and membrane lipids [8,185,186,187]. OS disturbs the homeostasis of RPE cells (including redox homeostasis) and their interactions with neighboring tissues and affects the majority of intracellular processes (phagocytosis, autophagy, metabolite transport, proliferation, cell death, etc.). High levels of ROS lead to the accumulation of oxidized lipoproteins, which inhibit the degradation of POS in RPE cells during phagocytosis [188]. In addition to lipids, ROS can also oxidize proteins, leading to the disruption of the proteostasis maintenance system. The main actors in the degradation of oxidized misfolded proteins are proteasomes [189]. Autophagy and proteasome deficiencies can cause protein misfiling in the ER, leading to long-term stress in the ER. These processes can disrupt the balance between the redox potential and calcium metabolism, which leads to the release of calcium from intracellular depots into the cytoplasm [3,155,190]. On the one hand, OS in RPE cells is an inducer of stress in the ER. On the other hand, ER stress directly supports the development of OS [191].

Ferroptosis, a new form of programmed cell death, was characterized by LPO and GSH depletion that was mediated by iron metabolism [192]. The molecular mechanisms underlying the interplay between OS and ferroptosis in RPE cells are the subject of active study. There is experimental evidence that the phenomenon of ferroptosis involves the disruption of the signaling pathways of genetic changes in iron homeostasis [193], the GSH metabolic pathway, and LPO metabolism [171].

Long-term stable changes in redox homeostasis that are influenced by exogenous or endogenous factors, in violation of interactions between RPE cells and adjacent tissues, can lead to the prevalence of oxidative processes and OS.

Chronic OS leads to global changes in RPE cell metabolism and disruption of the ADS [102,194]. A strong OS ultimately leads to the death of RPE cells and neurons [119].

The cascades of protective reactions are launched, which are mediated by calcium ions, ATP, and ROS released into the intercellular space from damaged cells [195,196]. When the RPE and photoreceptors are damaged or exposed to OS (H_2_O_2_), ATP secretion, activation of ATP-dependent Ca^2+^ channels (P2X7R), the release of Ca^2+^ from cell storage, and increased Ca^2+^ transport to the cell are promoted [197,198]. These molecular reactions modulate membrane permeability, promote mitochondrial swelling, and lead to cytochrome C release in the cytoplasm.

Transcriptional profiling showed an increased level of markers of apoptosis, autophagy, ER stress, and mitochondrial dysfunction in OS-dependent metabolic disorders in the RPE [199,200,201,202]. Ex vivo RPE cells from patients with AMD accumulated lipid droplets and glycogen particles, showed increased sensitivity to OS, contained destroyed mitochondria, showed reduced mitochondrial activity, and impaired autophagy, as determined by LC3-II/LC3-I markers [203].

### 5.2. The Role of Transcription Factors in RPE Response to OS

OS action is mediated by transcription factor-1 activator and transcription factors Fos and ATF, which control cell proliferation, autophagy, and apoptosis [125]. Transcription factors of the BCL-2, BAX, BAK, and BIM families control changes in mitochondrial membrane permeability, calcium release from the ER, and its entry into mitochondria [204]. An important role in the regulation of redox reactions in the RPE is played by apurine endonuclease-1, which is involved in the activation of activator protein-1, as well as that of transcription factors NF-κB and HIF-1α [204,205]. The choice of signaling pathways activated by OS is determined by the intensity and duration. Strong OS intensifies autophagy, while chronic OS depletes this process, as shown by the human RPE cell line ARPE-19 [9]. Low ROS concentrations induce signaling pathways involving transcription factors Nrf2 and Keap-1, the interaction between which regulates the operation of a large number of downstream genes [206,207].

OS also induces the expression of phase II enzymes: NADPH-quinine oxidoreductase-1, HO-1, the modifier subunit, and the catalytic subunit of glutamate-cysteine ligase [139,140]. ROS-producing NOXs are considered targets for the action of inhibitors of ROS formation, N-acetylcysteine (NAC), apocynin, and diphenylene iodonium [208,209,210]. However, average concentrations of ROS are mediated by transcription factors NF-kB and AP-1 and support inflammatory responses [128].

### 5.3. OS-Dependent Secretion of Growth Factors

The accumulation of oxidized phospholipids in the RPE stimulates the ATF4-dependent secretion of angiogenic factor VEGF, mediated by protein kinase CK2 [128]. Under conditions of OS, the secretion of growth factors by RPE cells is disrupted, which contributes to the cellular response of the RPE and can trigger signaling pathways in the RPE that mediate the development of angiogenesis and degenerative processes. It has been shown that OS induces the increased secretion of the transforming growth factor TGFβ in the RPE [211]. TGFβ signals and their associated effector Smad proteins, through de novo protein synthesis, enhance the secretion of angiogenic vascular growth factor VEGF in the RPE [45,212,213,214]. An increase in VEGF secretion was noted in non-polarized RPE cells with impaired intercellular tight junctions and ECMs under OS [215]. VEGF expression in the RPE is significantly increased by the synergistic action of TGFβ2 and TNFα, as well as that of TGFβ2 and thrombin [125,216,217,218]. Increased ROS production by mitochondria also contributes to the accumulation of the amyloid β protein and leads to the excessive secretion of VEGF [219].

An increase in the expression levels of VEGF and VEGFR receptors was found in exosomes isolated from the RPE after experimentally modeling OS [220]. Exosomes were suggested to play a role in the occurrence and development of choroidal neovascularization, where they are important in mechanisms of wet AMD pathogenesis and for developing therapeutic strategies for this disease [221]. An increase in the secretion of the isoform of the angiogenic factor VEGF-A in the RPE with a simultaneous decrease in the secretion of the angiogenic inhibitor PEDF is a key event in the pathogenesis of AMD [222,223]. The MAPK ERK1/2 pathway is known to be involved in the stimulation of VEGF secretion by RPE cells under OS [224]. TGFβ2-dependent activation of VEGF also involves the main MAP kinases (ERK1/2, p38) and a component of the TGFβ2 pathway, JNK [198,225]. An increase in VEGF secretion by RPE cells can also be mediated by activation of the Wnt/β-catenin signaling pathway [226].

### 5.4. Changes in ATP Metabolism

The concentration of ATP decreases inside the cells and increases outside the cells during OS [174]. High concentrations of extracellular ATP cause neuronal death, while maintaining the physiological level of adenosine is necessary for the functioning of the RPE and retinal neurons [175]. Enhanced production of ROS in the RPE, associated with the disruption of the integrity of cell membranes, increases the activity of calcium ATPase. Metabolic disorders, hyperglycemia, and OS can lead to profound changes in intracellular and extracellular nucleotide levels in the RPE and neurons [227]. A decrease in the intracellular concentration of ATP and an increase in the level of extracellular ATP concentration leads to the activation of purine receptors of the P2RX7 subtype [200]. ATP, an endogenous P2X7 receptor agonist, and BzATP, a synthetic P2X7 agonist, activate ATP-dependent Ca^2+^ channels. This induces a P2X7-mediated influx of extracellular Ca^2+^, an increase in its intracellular level, and an increase in the pH of lysosomes [228,229,230], which stimulates the release of interleukin-1β [231]. In turn, calcium activates oxidative enzymes (double oxidase-1 and NADPH-oxidase) and stimulates the production of hydrogen peroxide, which acts as a paracrine signal [232]. These processes contribute to the accumulation of LF in the RPE, which is a component of drusen, along with oxidized forms of lipoproteins [233].

### 5.5. Endoplasmic Reticulum Stress

With the accumulation of LPO products in the RPE from the ER, a reaction with a non-structured protein occurs with the aim of restoring homeostasis [234]. The ER maintains cellular calcium homeostasis through a complex set of calcium-dependent molecular chaperones required for protein folding [235]. The suppression of proteasome function inhibits the ER-associated protein degradation pathway, which enhances their misfiling in the ER and triggers ER stress [236]. Proteotoxic stress enhances OS, inflammation, and hypoxia. Proteasome inhibition induces ER stress and stimulates the expression of hypoxia-inducible factors (HIFs). HIFs regulate the expression of multiple growth factors and cytokines involved in angiogenesis and inflammation in retinal degeneration [237]. The stimulation of VEGF expression by transcription factor HIF-1α in human RPE cells has been shown [238]. The inhibition of proteasome degradation enhances the accumulation of LF [73]. ER stress induces an unfolded protein response (UPR) via the transducers IRE1 (inositol-requiring protein-1), PERK (protein kinase, RNA-like kinase ER), and ATF6 (activating transcription factor-6) [234]. Under ER stress, the activation of transducer proteins triggers signaling cascades that induce a downstream adaptive UPR via protein kinases and transcription factors. UPR signaling can stimulate the expression of molecular chaperones as well as antioxidants to restore cellular homeostasis. However, the insufficiency of this cell defense system can trigger a cell death program [239,240]. Recently, it was shown that the ER stress response under external OS was associated with the activity of Chac1, which is the component of the UPR pathway. Chac1 is known to be a candidate gene for the glutathione-metabolism signaling pathway that accelerates GSH degradation [241]. It has been suggested that Chac1 is involved in RPE ferroptosis signaling pathways as a downstream component of the ATF4-ATF3-CHOP cascade [242].

On the human ARPE-19 cell line, it was shown that the OS-inducer 7-ketocholesterol induces caspase-dependent apoptosis as a result of the use of receptor-mediated signaling pathways (with the participation of caspase-8 and caspase-12), which indicates the activation of ER stress. These processes are carried out without the participation of the mitochondrial pathway mediated by caspase-9 [243]. One explanation for the lack of caspase-9 activation may be the activation of HSPs, which are highly active in RPE cells [244]. It is assumed that HSPs can prevent caspase-9 cleavage during apoptosis initiation [245,246]. Caspase-12 is the main component of the caspase-dependent signaling pathway induced by ER stress. Inactive caspase-12 is localized on the cytosolic surface of the ER; however, upon OS activation, it can trigger a cascade of reactions leading to the activation of effector caspase-3 [247].

### 5.6. Reorganization of the Cytoskeleton

The role of nitric oxide in the modulation of cytoskeletal reorganization was demonstrated in the model of retinal degeneration 1 (rd1). Proteomic analysis showed that the expression levels of vimentin and serine/threonine protein phosphatase 2A (PP2A) are significantly increased when mice are exposed to continuous light exposure for 7 days compared to 12 h light/dark cycling conditions. Simultaneously, nitric oxide inactivates the PP2A catalytic subunit, which leads to increased phosphorylation of vimentin, which is a substrate for this phosphatase [248]. OS-mediated accumulation of Aβ induces inflammatory activity, oxidative phosphorylation dysregulation, angiogenesis, and cytoskeleton destabilization, causing a large amount of damage in the subretinal region, which is associated with the pathogenesis of AMD [219,249]. Aβ dislocates occludin and decreases the levels of occludin and zonula occludens-1 mRNA expression in RPE cells. This molecular alteration destabilizes the link between the transmembrane and the actin cytoskeleton and influences the transepithelial permeability of RPE cells [250].

### 5.7. Communication between Mitochondria and Lysosomes in the RPE Cellular Response to OS

Molecular processes in the RPE working against OS are associated with the dysregulation of mitochondrial function. They lead to a decrease in the activity of glyceraldehyde-3-phosphate dehydrogenase and the accumulation of advanced glycation end products and polyols [112,251,252]. High concentrations of ROS cause the destruction of mitochondrial membranes and the disruption of membrane transporters, which leads to apoptosis, necrosis, or ferroptosis [13,253]. Depolarization of mitochondrial membranes and the accumulation of LPO products (acrolein, etc.) further stimulate ROS production by damaged mitochondria [254,255]. Data have been obtained that show that the damage to RPE cells in AMD caused by the accumulation of LPO products is mediated by stimulation of the MAPK cascade [256]. There is cross-talk between mitochondria and lysosomes during the OS-induced apoptosis of RPE cells. Released lysosomal enzymes permeabilize mitochondrial membranes, which enhances ROS production and the process of lysosome membrane permeabilization (PML) [114]. PML is considered one of the early events in many cases of apoptotic cell death. This process is mediated by the action of cathepsin proteases released from lysosomes involved in the caspase-independent signaling pathway of apoptosis in RPE cells [257,258]. If PML is not a trigger event for apoptosis, this process is induced at later stages by other mechanisms that enhance signals for cell death [259].

### 5.8. Apoptosis Signals

OS in the RPE and photoreceptors disturbs the signaling pathways of antioxidant defense and triggers signaling pathways of cell death in a certain form (apoptosis, necrosis, or toxic cell damage), which allows for particular levels of OS severity and cell destruction [94,119].

Mitochondrial calcium overload leads to swelling and can subsequently cause an outflow of apoptogenic cytochrome c factors and apoptosis-inducing factors from the mitochondria into the cytoplasm, where they activate caspase cascades and ultimately lead to apoptosis of the RPE and neurons [258,260]. OS triggers key apoptosis signaling in RPE cells via c-Jun N-terminal kinase (JNK)/stress-activated protein kinase, p38 and ASK1, NF-κB signaling pathways, and mitochondrial activation of the PKC signaling pathway [112,194,261]. NF-κB stimulates excessive production of the pro-inflammatory interleukin IL-8 and HIF-1 [262]. An in vitro system was used to show that human RPE cells treated with hydrogen peroxide for 24 h respond with increased production of pro-inflammatory cytokines NF-κB and IL-6 and the phosphorylation of p38, MAPK, ERK, JNK, and intercellular adhesion molecule 1 (ICAM- 1) [263]. Thus, the model of CoCl_2_-induced hypoxia showed a connection with the molecular processes of OS development in human RPE cells and was accompanied by a significant increase in LPO, the accumulation of NF-κB transcription factors in the perinuclear space [264], an increase in the secretion of VEGF, and the activation of caspase-3 and poly (ADP-ribose)-polymerase, which triggers signaling pathways for cell apoptosis [238,265,266]. Caspases and other components of the ROS-induced pro-apoptotic cascade are promising therapeutic targets for the development of apoptosis in the RPE and degenerative pathologies associated with its dysfunction [267].

### 5.9. The Influence of OS on the State of Chromatin in RPE

OS affects the activity of epigenetic proteins and chromatin remodeling. Currently, attempts are being made to map associated epigenetic and transcriptional changes in the RPE in normal and pathological conditions (AMD, PVR, retinitis pigmentosa, etc.) [268,269]. It has been shown that epigenomic changes associated with the development of these pathologies affect the functioning of transcription factors. Epigenomic changes lead to an increase in active chromatin (euchromatin) marks on the regulatory sites of DNA enhancers of putative targets for binding to transcription factors [270]. The OS-induced formation of inactive chromatin (heterochromatin) in mammalian RPE can be considered an important link in the mechanisms of protection of these cells from oxidative damage. OS-induced selective accumulation of heterochromatin near the promoters of the p53 tumor suppressor gene, which undergoes desumoylation, a process that facilitates the interaction of p53 with heterochromatin, was revealed, as shown in in vivo and in vitro systems. Epigenetic changes and chromatin conformation lead to locus-specific transcription recruitment and repression, serving as a mechanism for regulating RPE cell survival [271].

## 6. The Exogenous Regulation of Redox Homeostasis of RPE Cells

At present, neuroprotection strategies aimed at mobilizing endogenous defense systems and enhancing the antioxidant defense of RPE cells remain the most effective [272,273,274,275]. The RPE is closely associated with the neural part of the retina; therefore, the restoration of photoreceptor functions protects the RPE from OS and vice versa, where the protection of the RPE from OS prevents degenerative changes in the retina. The RPE and photoreceptors are key cellular targets in the treatment of OS-dependent pathologies of the RPE and retina, among which AMD occupies a special place. As noted, cellular events in the pathogenesis of AMD are ER stress, decreased autophagy and proteasome activity, mitochondrial dysfunction, and the activation of cell death signaling pathways. Molecular participants in these processes are promising targets for the development of new strategies to stabilize RPE homeostasis and prevent retinal degeneration [276,277,278,279]. The genetic aspects of AMD, as well as the use of animal models to develop approaches for the potential treatment of AMD in humans, have been reviewed in recent reviews [280,281,282].

### 6.1. The Aging of RPE Cells as an Endogenous Factor in the Development of Retinal Pathologies

The RPE displays several specialized functions essential for retinal homeostasis (see Figure 1). The disturbed structure and dysfunction of an aging RPE lead to degenerative retinal diseases, such as AMD. AMD primarily affects RPE cells with the subsequent degeneration of photoreceptors. OS is considered to be a major AMD risk factor. The aging of RPE cells is related to a progressive decline in their functions that can be coupled with ROS overproduction [283].

The RPE undergoes several structural changes during aging. These include the loss of melanin, the formation of drusen, the thickening of BM, the accumulation of LF, decreased mitochondrial mass, a disturbed mitochondrial network, and microvilli atrophy [284]. Cellular senescence, initially an irreversible inhibition of cellular division, is associated with a decline in basic cellular functions, such as differentiation, phagocytosis, and autophagy. Senescence-associated secretory phenotypes (SASPs) are associated with the release of ROS, selective growth factors, and inflammatory cytokines, chemokines, and proteases [285].

An age-dependent phagocytosis activity reduction may not only disrupt retinal homeostasis but also can affect RPE survival, leading to RPE apoptotic loss and photoreceptor degeneration. An age-related decrease in lysosomal enzymatic activity inhibits the autophagic clearance of outer segments, mitochondria, and protein aggregates, thereby accelerating the accumulation of LF and products of LPO in RPE cells [286]. Decreased autophagy is associated with the cell phenotype shifting to senescent cells that contribute to a loss of tissue homeostasis. The accumulation of LF in the RPE is a sign of senescence in AMD [287]. The exposure of LF and fatty acids to light initiates a production of ROS that may damage mitochondria and mitochondrial DNA [288]. Deposits of amyloid beta (Aβ) may also play an important role in AMD pathogenesis. These deposits may be associated with several consistent factors of AMD pathogenesis, including OS, a decline in mitochondrial and lysosomal functions, inflammation, and certain genotypes of the complement system. Aβ was shown to induce senescence of RPE cells and impair mitochondrial metabolism [289,290].

The contents of antioxidant proteins and small-molecular-weight antioxidants decline in an aging RPE [291]. An age-related decrease in the activity of many proteins pivotal to the antioxidant system has been observed, including Nrf2, SOD1-2, CAT, and glutathione peroxidase (GPX) [292,293,294]. Additionally, DNA repair, both in mitochondria and the nucleus, decreases with age [295]. The expression of UCP2, a proton transport protein in the inner membrane of the mitochondria, in ARPE-19 and RPE cells from younger donors was higher than in those from older donors. In conclusion, the expression of the *UCP2* gene was decreased in aged RPE cells, promoting the lower ability of antioxidants in these cells [296].

The proteomic profile of RPE cells in AMD was distinct from that observed with aging. For example, in contrast to the decrease in glycolysis observed with aging, glycolytic enzymes were increased with AMD. This metabolic switch that favors glycolysis is likely due to defects in mitochondrial function. There were also multiple indicators of elevated OS and the activation of inflammation and immune response [297]. Thus, AMD should be considered a complex disease with an interplay between aging and other risk factors such as OS, hypertension, and diabetes.

### 6.2. Strategies for the Exogenous Protection of RPE against Oxidative Stress

In this part of the review, special attention is paid to possible approaches for the treatment of AMD that target ROS acceptors, reduce the risk of developing OS and its consequences, reduce the progression of AMD, and maintain the metabolism of border tissues (photoreceptors and choroid tissues) [298,299]. The developed strategies for protecting the RPE from OS involve the use of inhibitors of intracellular ROS that influence key links in the mechanisms of antioxidant protection, including stress-sensitive transcription factors and their targets. Other approaches to counteract OS in the RPE are based on the activation of signaling pathways that involve the key redox-dependent transcription factor Nrf2 [121,122,124]. This transcription factor controls almost all major components of the antioxidant defense of RPE cells, and it is thought to remodel chromatin structures and facilitate the formation of the general transcription machinery via its secondary and tertiary coactivators [123].

Currently, the mechanisms of action of the pharmacological agents of protection against OS continue to be actively studied. We analyzed the main strategies from among the most promising approaches to prevent OS and overcome its consequences in RPE cells. We systematized these approaches according to the principle of their action (Figure 3).

#### 6.2.1. Direct Neutralization of ROS with the Help of Exogenous Antioxidants

Putative exogenous OS protectors include resveratrol, curcumin, acetylcysteine, multivitamins, metal oxide nanoparticles, polyphenolic compounds, spermidine, nucleosides, agonists of serotonin receptors, and agonists and antagonists of the purinergic system [298,299,300]. An example is the use of spermidine, a free-radical scavenger that inhibits the degradation of singlet oxygen and reduces the production of ROS and reactive nitrogen species, mainly through the activation of MAPK, ASK-1, and p38 [301,302]. Studies have confirmed that some natural antioxidant compounds can act as ROS scavengers, enhance antioxidant enzymes, and induce or inhibit signaling pathways and gene expression related to stress response and cell death. The most common fat-soluble antioxidants include α-tocopherol (vitamin E) and carotenoids (β-carotene, lutein, and zeaxanthin). These biomolecules, being lipophilic, are predominantly associated with the membranes of lysosomes and mitochondria. The action of α-tocopherol, which is one of the most powerful antioxidants, is based on its ability to quickly interrupt the autocatalytic LPO reaction [170,171,303]. A similar mechanism underlies the action of carotenoids, which are capable of scavenging peroxyl radicals or neutralizing singlet oxygen [170,304,305]. Under in vitro conditions, spermidine suppresses OS in RPE cells by blocking an increase in the level of intracellular calcium [300]. A pharmacological agent from the group of iron chelators, deferiprone, was used in a model of iron-induced degeneration of the mouse retina, which manifested in a decrease in OS and the prevention of neuronal degeneration [306]. It has also been shown in RPE cell lines that the addition of iron chelators makes these cells less sensitive to H_2_O_2_, delays the accumulation of oxidative damage, and stabilizes cell functions [307].

#### 6.2.2. The Suppression of ROS Production via the Decreased Expression of Pro-Oxidant Genes and the Activity of Pro-Oxidant Enzymes

Among the biologically active antioxidant molecules, much attention has been focused on studying the mechanisms of action of red wine polyphenol resveratrol, as evidenced by a large number of studies. Using different models in vivo and in vitro, many aspects of the action of resveratrol have been identified, which are illustrated in this section of the review. Using a model of ultraviolet A (400–315 nm)-induced OS, it was shown that resveratrol is able to suppress the production of H_2_O_2_ in RPE cells by the intracellular activation of p38 and kinase, which increases the viability of these cells [308]. An important aspect of the action of resveratrol is the weakening of ROS production in POSs, as well as the neutralization of the toxic effect of A2E in human RPE cells [309].

Resveratrol has been shown to impact several transcription factors (AP-1 and Egr-1), cell cycle regulators (p21Cip1/WAF1), and apoptosis (p53, Bcl-2, Bax, and survivin) [310]. Resveratrol suppresses hypoxia-inducible factor-1α accumulation and vascular endothelial growth factor (VEGF) secretion, while in endothelial cells, it inhibits VEGF-R2 phosphorylation, suggesting a role of resveratrol in the inhibition of angiogenesis and choroidal neovascularization [311]. Resveratrol is a potent inhibitor of multiple signaling pathways related to fibrosis development, e.g., TGFβ/SMAD, NF-κB signaling, and ERK signaling pathways [312]. Resveratrol not only protects ARPE-19 cells from H_2_O_2_-induced death by inhibiting oxidation processes but also prevents the migration and hyperproliferation of RPE cells by activating the phosphatidylinositol-3 kinase/Akt pathway and MAPK)/ERK 1/2 cascade [313,314]. The RPE cell line ARPE19 model showed that its inhibitory effect is mediated by the suppression of PDGF receptor β, phosphatidylinositol-3 kinase/Akt pathway activation, and the MAPK signaling cascade [314]. Resveratrol inhibits the TGFβ2-induced EMT of RPE cells by deacetylating SMAD4 and targeting the EMT of the RPE in a SIRT1-dependent fashion. The deacetylation of Smad4 leads to the suppression of cell proliferation and migration, as well as the suppression of the excessive synthesis of ECM components (fibronectin) in the RPE [315]. It was shown that pretreatment with crocetin for ARPE19-stressed cells positively influenced metabolic functions, including mitochondrial respiration and glycolytic instigation, as well as membrane integrity. This action is realized via the activation of the extracellular signal-regulated kinase (ERK) pathway, a member of the MAPK signaling cascade. Furthermore, this crocetin-induced protection is comparable with that from well-known antioxidants, such as vitamin C or E [316].

These data expand our understanding of the targets and mechanisms of action of resveratrol in the treatment of AMD and prevention of PVR. It is hypothesized that resveratrol could be a potential therapeutic application for PVR due to its suppressive effect on the fibrotic process that prevents the development of PVR.

Other authors have shown that resveratrol inhibits Akt/protein kinase activity and activates p53 in the choroid endothelium in vivo [317]. Resveratrol inhibits macrophage infiltration and inflammatory and angiogenic cytokines in the RPE–choroidal complex, including VEGF, intercellular adhesion molecule-1 (ICAM-1), and COX-2 [298,318]. Resveratrol has the ability to suppress the production of ROS in RPE cells in a model of atrophic AMD caused by the toxic effects of sodium iodate [319], and the secretion of pro-inflammatory chemokines IL-6, IL-8, and monocyte chemoattractant protein (MCP)-1 [319,320]. Resveratrol maintains the viability of RPE cells exposed to H_2_O_2_ based on the modulation of superoxide dismutase/malonic dialdehyde activity, the activation of Bcl-2, and the suppression of the expression of cleaved caspase-3, which leads to the inhibition of apoptosis [321].

Pharmacological inhibitors of ROS, NAC, apocynin, and diphenylene iodonium prevent the excessive production of VEGF by human choroid endothelial cells and the production of ROS, the source of which is NOX [208,209,210]. NAC reduces the production of the angiogenic growth factor VEGF-A isoform and the expression of caspase-3 and CHOP, preventing the death of ARPE-19 cells in a model of degeneration caused by exposure to retinoic acid [267]. RPE cell exposure to NAC or thioredoxin was shown to effectively inhibit ASK1 signaling and prevent apoptosis in the RPE [322] via the inhibition of ASK1 kinase and NF-κB transcription factor [323]. Data obtained on different models allow us to consider NAC as an effective molecular tool in the treatment of AMD [324], maintaining the stability of RPE cell homeostasis, and the treatment of retinal degenerative processes in retinitis pigmentosa [325].

The inhibitor of apurine endonuclease-1 E3330 prevents the accumulation of NF-κB in the nucleus and the secretion of chemoattractant protein-1 in monocytes, reducing retino-choroidal angiogenesis and the death of retinal neurons induced by laser irradiation. The action of E3330 in the RPE is realized by reducing the expression of stress-sensitive transcription factors (Nrf1/Nrf2, p53, nuclear factor NF-κB, p65, HIF1, CBF/NF-Y/YY1, and MTF-1). E3330 prevents the accumulation of NF-κB in the nucleus, the secretion of chemoattractant protein-1 by the monocyte MCP-1, and the excessive production of VEGF [205].

The pharmacological agent epigallocatechin-3-gallate protects human RPE cells from death and increases their viability as a result of the inhibition of ROS production, which leads to a decrease in the permeability of the endothelium of the microvascular bed of the eye, preventing pathological remodeling of the RPE. This effect is mediated by a decrease in the expression of pro-angiogenic factors including VEGF, VEGF-2 receptor, and matrix metalloproteinase-9 (MMP-9) [326,327]. Other authors have shown that activation of the ERK/CREB signaling pathway, MMP-14 and TIMP2 metalloproteinases, and Toll-like receptor 3 (TLR3) protects the RPE from the toxic effects of ROS (hydrogen peroxide) [328,329,330,331]. Melatonin plays an important role as an antioxidant and antiproliferative factor in the endogenous protection of RPE cells. Melatonin synthesized in RPE cells paracrinely has a positive effect on photoreceptors by inhibiting the synthesis of transcription factor HIF-1α (hypoxia-inducible factor-1α) and the excessive secretion of VEGF [332]. In addition to VEGF, HIF-1α controls genes that are involved in the mechanisms of antioxidant defense and neuroprotection [333].

Under the action of melatonin, which has been constantly tested with experimental animals (mice), a return to the normal state of the cytoskeleton and, in particular, the proteins of intermediate filaments vimentin and PP2A was observed, which stabilizes the state of the ECM [248]. The role of the exogenous precursor of melatonin (N-acetyl-5-methoxytryptamine) in the modulation of the RPE and retinal functions was revealed, including reducing inflammation and apoptosis, restoring the integrity of the external HRH, and reducing the level of VEGF and nitric oxide under OS conditions. The proven absence of side effects of melatonin on the RPE and retinal cells, even at high doses, made it possible to propose its use as a promising pharmacological agent to prevent or slow down the progression of AMD [334,335]. The combined use of melatonin and memantine demonstrated the effectiveness of their protective action against apoptosis and mitochondrial dysfunction in ARPE-19 cells in vitro after OS induced by ethylpyridine (a component of cigarette smoke). This effect is realized as a result of a decrease in the production of ROS and VEGF, the activity of caspase-3 and -9, and the activity of procaspase and poly (ADP-ribose) polymerase [336]. The regulation of the BM state underlies experimental approaches aimed at stabilizing intercellular interactions between the RPE and photoreceptors [275]. The use of drugs that reduce the level of oxidized forms of cholesterol (oxysterols) in RP can prevent the development of AMD [243,337,338,339]. The addition of exogenous neuprotectin D1 to cultured human RPE cells protected the cells from OS damage as a result of the activation of the expression of anti-apoptotic proteins Bcl-2 and Bcl-xL due to a decrease in the expression of pro-apoptotic markers Bax and Bad [159,160]. Neuroprotectin D1 inhibits the expression of caspase-3 and cyclooxygenase-2, which are activated during OS and act through the PI3K/Akt signaling pathway (phosphatidylinositol-3-kinase/Akt kinase) [164].

The use of thioredoxins is aimed at inhibiting the perinuclear accumulation of the nuclear transcription factor NF-κB, depolarization of the mitochondrial membrane, and activation of apoptosis signal-regulating kinase 1 (ASK1), which manifests itself through a decrease in RPE cell death [323]. Previously, it was demonstrated that the treatment of RPE cells with thioredoxin and/or NAC effectively inhibits the ASK1 signaling pathway, preventing the development of apoptosis [322].

#### 6.2.3. The Induction of Antioxidant Gene Expression and the Activation of their Enzymatic Products

Nrf2 activation as well as NF-κB inhibition are the main ways to prevent ROS-induced processes in the RPE and photoreceptors [340,341]. Strategies aimed at activating the components of the ADS are associated with the use of exogenous carotenoids. Thus, the use of astaxanthin can significantly reduce the formation of ROS induced by hydrogen peroxide and the apoptosis of RPE cells by activating the PI3K/Akt signaling pathway and enhancing the expression of the transcription factor Nrf2 [165]. An increase in Nrf2 activity, as well as the inhibition of NF-kB, showed promise for preventing ROS-induced events in the RPE and photoreceptors in AMD [340,341].

Lutein and vitamin E reduce UV-induced ROS production and LPO and increase the activity of antioxidant enzyme activities, which reduces apoptosis and increases RPE cells’ viability [342,343]. Lycopene inhibits NF-κB expression in the RPE, which is largely due to the increased expression of Nrf2 and GSH and the decreased expression of ICAM-1, having an antioxidant effect [171]. Previously, it was suggested that daily prophylactic administration of zeaxanthin can slow the atrophy of the RPE and photoreceptors, reducing the risk of the development and progression of AMD. These degenerative processes are characteristic of age-related macular degeneration in AMD [310]. The mechanism of action of zeaxanthin has been established, which consists of protecting RPE mitochondria from oxidative damage. Zeoxanthin acts by inducing Nrf2-dependent antioxidant enzymes, thereby maintaining the RPE structure and function [344]. RPE protective properties have been found for the drug α-mangostin, which enhances the antioxidant activities of Nrf2 and its target *HO-1* [275,345].

The mechanisms of the antioxidant action of fullerenol, a polyhydroxylated derivative of fullerene, are based on a decrease in ROS production and *p53* and *p21* gene expression, as well as the activation of sirtuin 1- and Nrf2-controlled genes of the ADS [346,347].

15-deoxy-∆12,14-prostaglandin J2 activates GSH synthesis, protecting RPE cells from oxidative damage [348]. Diarylheptanoids isolated from turmeric (*Curcuma comosa*) attenuate an H_2_O_2_-induced decrease in the activity of GPX and SOD, meanwhile, they increase the activity of catalases, reducing RPE apoptosis [349,350].

#### 6.2.4. The Activation of Autophagy

The action of autophagy is tightly linked to the regulation of redox homeostasis and seems particularly promising to prevent OS in the RPE. The possibility of modulating RPE redox homeostasis via the pharmacological action of the azapeptide ligand MPE-001 on CD36 (differentiation cluster 36) has been shown. MPE-001 has antioxidant and anti-apoptotic effects on RPE cells as a result of stimulation of the autophagy process, maintaining the viability of RPE cells and retinal photoreceptors [11]. In ARPE-19 cells treated with 5-(N,N-hexamethylene)amiloride (HMA), an inhibitor of pH regulators and Na^+^/H^+^ ion exchange, autophagy activation was observed under conditions of moderate OS [42]. One of the approaches to autophagy induction and RPE cell protection relies on inhibitors of peroxisomes and the PI3K-Akt-mTOR signaling pathway [75]. Independent studies on the ARPE-19 cell line have shown that resveratrol participates in the induction of autophagy and the formation of autolysosomes by modulating the expression of the *p62* and *LC3* genes [351,352].

#### 6.2.5. Redox-Sensitive MicroRNA

miRNAs are involved in the regulation of redox homeostasis in the RPE and border tissues, which makes them effective targets in the treatment of OS-associated pathologies [353]. Redox-sensitive miRNAs play an important role in the regulation of antioxidant signaling pathways in the RPE. Recent studies have shown that OS enhances the expression of miR-144-3p and mir-144-5p, which block the expression of Nrf2 and controlled target genes for antioxidants Nqo1 and Gclc, reducing the content of GSH in human and mouse RPE and increasing cell death [354,355]. *Nrf2* is a target gene of miR-93; the overexpression of Nrf2 alleviates the high glucose-induced apoptotic effect of ARPE-19 cells and can reverse the pro-apoptotic effect and inflammation of miR-93 [356].

The positive effect of microRNAs on the RPE realized by the regulation of NOX4 activity has been demonstrated previously. This effect is accompanied by a reduction in VEGF-induced ROS synthesis by RPE cells, neovascularization, and the EMT of RPE cells [357]. MiR-146a, activated in the RPE and choroid during cellular senescence, decreases VEGF activity in RPE cells [358,359]. MiR-93 and miR-302d inhibit the TGFβ signaling pathway and can be considered as a potential approach to prevent TGFβ2-mediated RPE conversion in mesenchymal transformation [360].

#### 6.2.6. Synthetic Peptides and Blockers of ROS and Neoangiogenesis

The key role of α1 Na^+^/K^+^-ATPase (α1 NKA) in the regulation of OS and the Nrf2 signaling pathway in the retina of mice with oxygen-induced retinopathy has been demonstrated. The α1 NKA targeting strategy using the pNaKtide peptide blocks the formation of α1 NKA/Src/ROS amplification loops and restores physiological ROS values. Strategies to decrease excessive VEGF production have been proposed using α1 NKA as the key target. Unlike other anti-VEGF pharmacological agents, pNaKtide inhibits inflammatory responses and retinal neovascularization [84]. The use of antioxidants that inhibit the activity of SMAD2, SMAD3, and VEGFA proteins and stabilize BM [275] by reducing the level of oxidized forms of cholesterol (oxysterols) in the RPE is considered a therapeutic approach to blocking uncontrolled angiogenesis, maintaining the stability of RPE cell homeostasis, and preventing cell death [284,359].

Another neuroprotective strategy assumes reducing excessive VEGF production by suppressing amyloid-β aggregate accumulation to prevent retinal toxicity in the retina in AMD [361]. The synthetic peptide inhibitor MRZ-99030 was shown to significantly reduce the number of apoptotic cells and attenuate amyloid-β toxicity via the formation of amyloid-β1-42 assemblies that are benign to the retina. Its unique mechanism of action, good bioavailability, and tolerability were observed in in vitro and in vivo models [362,363].

#### 6.2.7. Components of the Purinergic Signaling Cascade

In addition to the considered approaches aimed at activating the key components of antioxidant mechanisms in RPE cells, other methods of direct or indirect regulation of OS-dependent signaling pathways continue to be developed.

The activation of AMP-activated protein kinase (AMPK) or the inhibition of adenosine kinase can be used to protect the RPE from OS [364]. An increase in the level of cytoplasmic cAMP restores the acidic pH of lysosomes as a result of the stimulation of adenosine A2A receptors (A2AR) by the agonist adenosine [365,366]. Data strongly suggest that cAMP signaling can reduce inflammatory mediators in multiple retinal cell types, which protects the retina against stressors [367]. In summary, the elevation of adenosine signaling represents the positive response of RPE cells to AMD. Adenosine stimulates A2AR and reacidifies lysosomes. The mechanism underlying the restoration of the acidic lysosomal pH may be linked to the elevation of cytoplasmic cAMP following the stimulation of A2AR [230]. The use of the P2Y2 receptor agonist INS37217 promotes the restoration of the RPE transport function, photoreceptors, and cell homeostasis [368].

#### 6.2.8. Genome Editing

Genome editing technologies have revolutionized the gene therapy approach that is commonly used to deliver an exogenous transgene to cure a monogenic disorder. CRISPR/Cas9-mediated gene editing has successfully corrected mutations associated with various ophthalmic pathologies such as AMD, retinitis pigmentosa, and Leber congenital amaurosis [369]. Additional possibilities have opened up in the field of gene and cell therapy for retinal degeneration diseases through a combination of the CRISPR system with induced pluripotent stem cells (iPSCs), especially in the late stages with the irreversible loss of the RPE and photoreceptors [370]. The nearly limitless supply of RPE iPSCs provides the opportunity for both large-scale drug testing/screening platforms and patient-specific testing [371], and for investigating the mechanisms of regulation of cellular homeostasis and disease modeling [372]. The strategies that have an impact at the level of the downregulated components of signaling cascades and that are involved in the control of cell processes have the advantage because they allow us to prevent the inhibition of all signal transmission functions. At the same time, the underlying targets can bind the different cross-talk signaling pathways. In this regard, the search for the specific components and targets of OS-dependent signaling pathways remains relevant.

## 7. Conclusions and Perspectives

RPE cells are the key targets for protection against OS and the prevention of RPE-related neurodegenerative pathologies. The genetic aspects of these abnormalities and the problems of using model animals to develop RPE therapies in humans have been recently reviewed elsewhere [281,282,284]. Analysis of the available data makes it possible to propose strategies to recover RPE function. To date, the most efficient approaches rely on the mobilization of endogenous defense systems and enhancing antioxidant protection in RPE cells [272,273,274,275]. In the context of OS-dependent neurodegenerative diseases, which depend on the state of the RPE cell layer, an imbalance in signals in the autophagy system and the activity of molecular chaperones in the RPE can significantly aggravate the situation due to strong OS. In this regard, the impact (activation) of the autophagy system in the RPE and retinal photoreceptors is considered one of the most promising approaches [373]. Possible effective approaches to RPE cell stabilization focus on ROS acceptors, lowering the risk of OS and progressive retinal degeneration, and metabolism maintenance in the neighboring structures and tissues (BM, photoreceptors, and the choroid). The main focus has shifted to antioxidants, which simultaneously decrease ROS levels and decrease or suppress the expression of the relevant pro-oxidant and inflammation genes. The main methods for maintaining RPE functions and retinal homeostasis are aimed at blocking the activity of pro-oxidant biomolecules, the consequences of the negative influence of OS factors, and/or the activation of endogenous protective resources. The development of genomic, transcriptomic, and epigenomic technologies has provided researchers with powerful tools for identifying and analyzing the activity and interactions of molecular targets, including tissue-specific regulators of cellular processes in the RPE [374]. Despite the progress that has been made in studies of RPE abnormalities (AMD), there are still no efficient methods to stabilize the RPE state and arrest early degenerative processes. To a large extent, this is due to the multifactorial nature of the abnormalities [143]. Most of the work on the effect of OS on the functioning of cell defense systems has been performed on in vivo animal models [97,101,375,376,377]. A number of functions of regulatory factors have been largely determined by studying the effects of various antioxidants on in vitro RPE cell systems [378]. Problems of transferring the accumulated knowledge from the level of a cell to the scale of a whole multicellular organism, as well as identifying common patterns for mammals in the functioning of the antioxidant system, still exist. Another problem is the extensive cross-talking between producers and mediators of OS and the signaling pathways that control the cell response and its impact on OS and antioxidants. As shown through the analysis of the data accumulated, molecular participants in different parts of the ADS of the RPE can trigger or modulate multiple signals. The functioning of signaling pathways largely depends on the intensity and duration of OS and on the state of RPE cells and demonstrates the possible use of alternative signal transduction pathways. This makes it difficult to target the signaling to modulate cellular processes, as most of the molecular targets and signaling pathways remain largely obscure. In this regard, controversial questions regarding insufficient selectivity and side effects of the drugs for neuroprotection remain. The experimental impact on molecular targets for blocking the production of ROS or activating an endogenic defense requires a comprehensive knowledge of how to finetune their participation in the regulation of the vital functions of RPE cells.

Understanding the entire spectrum of functions of redox homeostasis regulators in the RPE and photoreceptors and their response to OS, as well as taking into account the specificity of their metabolic pathways, is the key to effective antioxidant therapy. Further studies of all levels of ADS regulation in RPE cells will provide a better understanding of the possibilities for therapeutic intervention (neuroprotection).

## Figures and Tables

**Figure 1 ijms-24-10776-f001:**
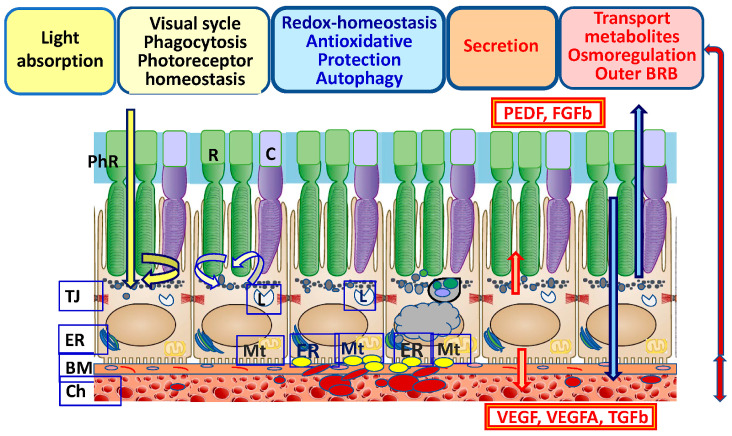
Structure and physiological functions of RPE cells that ensure the maintenance of their homeostasis. Abbreviations: TJ—tight junctions; BM—Bruch’s membrane; Cho—choroid; PhR—photoreceptors; R—rod; C—cone; ER—endoplasmic reticulum; Mt—mitochondria; L—lysosome; BRB—blood–retinal barrier; PEDF—pigment epithelium-derived factor; FGFb—basic fibroblast growth factor; VEGF—vascular endothelial growth factor; TGFb—transforming growth factor-beta.

**Figure 2 ijms-24-10776-f002:**
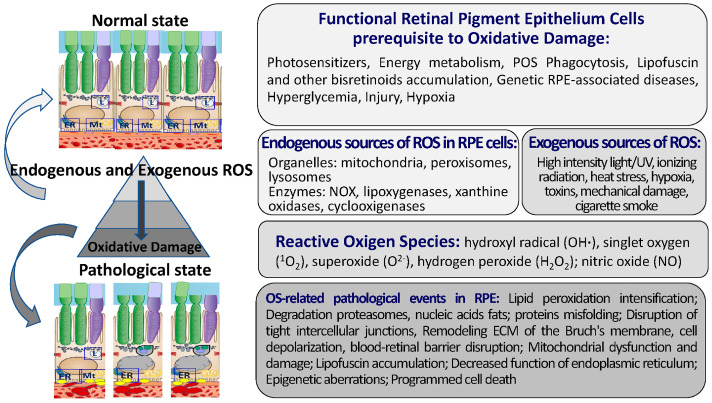
Development of pathological processes in RPE under oxidative stress conditions. RPE cells are characterized by high metabolic activity and a predisposition for oxidative damage. Imbalance of redox homeostasis of RPE cells under endogenous and exogenous sources of ROS leads to the development pathological processes and cell death. Abbreviations: POSs—photoreceptor outer segment fragments; RPE—retinal pigment epithelium; ECM—extracellular matrix; ER—endoplasmic reticulum.

**Figure 3 ijms-24-10776-f003:**
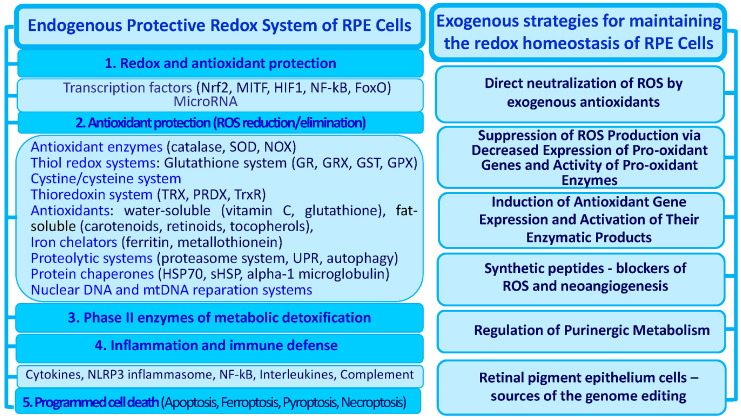
Levels of the RPE cells’ defense system regulation against oxidative stress and antioxidant neuroprotective strategies. Strategies of endogenous (left) and exogenous (right) protection of RPE cells against oxidative destruction are shown. Abbreviations: RPE—retinal pigment epithelium; ROS—reactive oxygen species; GCLM—glutamate-cysteine ligase modifier subunit; GCLC—glutamate-cysteine ligase catalytic subunit; GR—glutathione reductase; GST—glutathione S-transferases; GRX—glutaredoxin; GPX—glutathione peroxidase; HO-1—heme-oxygenase-1; NOX—NADPH oxidases; NQO1—NADPH-quinine oxidoreductase-1; PRDX—peroxiredoxin; SOD—superoxide dismutase; TRX—thioredoxin; TrxR—thioredoxin reductase.

## Data Availability

Not applicable.

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
