# Peer review of "Endogenous and Exogenous Regulation of Redox Homeostasis in Retinal Pigment Epithelium Cells: An Updated Antioxidant Perspective"

_ijms, 2023, doi:10.3390/ijms241310776_

Round 1

Reviewer 1 Report

This is a well written review that meets a high academic standard.  I have only a few comments that I feel would improve the review for a broad readership.

1)    The authors mention the production of ROS by mitochondria but do not mention the mechanisms intrinsic to mitochondria that can regulate ROS through controlling the mitochondrial membrane potential (MMP).  There is a substantial body of literature on the role of uncoupling proteins in controlling MMP, as well as the role of several other carrier molecules.  While not all of this literature is RPE-related, there is enough to draw some general conclusions about these molecules in RPE.

2)    Related to the above, can the authors estimate the relative contributions of ROS from the environment versus those produced form the high metabolic flux through the lectron transport chain?

3)    On p9 l369-372 the authors mention an interaction between Nrf2 and Mitf but only say “Interactions between these transcription factors control melanogenesis, mitochondrial biogenesis, and redox homeostasis [135].” It would greatly help the reader to be told what this interaction consists of and how it actually controls the listed functions.  More depth is always helpful in an article of this type.

4)    Though mentioned briefly, and a number of references discuss it, the whole topic of RPE and aging is not dealt with adequately.  A separate section summarizing what we know about RPE and aging, and its relationship with diseases such as ARMD, would be very valuable.

Author Response

Response to Reviewer 1 Comments

General comment:  This is a well written review that meets a high academic standard.  I have only a few comments that I feel would improve the review for a broad readership.

Specific comments:

Point 1: The authors mention the production of ROS by mitochondria but do not mention the mechanisms intrinsic to mitochondria that can regulate ROS through controlling the mitochondrial membrane potential (MMP).  There is a substantial body of literature on the role of uncoupling proteins in controlling MMP, as well as the role of several other carrier molecules.  While not all of this literature is RPE-related, there is enough to draw some general conclusions about these molecules in RPE.

 Response 1: There are some mitochondrial carriers expressed in RPE which are regulated MMP under stress conditions. In addition to UCP2, 2-oxoglutarate carrier and dicarboxylate carrier were recently identified in RPE which transport glutathione into mitochondria. We added corresponding remark in section 4.1.1 on page 7.

Point 2: Related to the above, can the authors estimate the relative contributions of ROS from the environment versus those produced form the high metabolic flux through the electron transport chain?

Response 2: The relative contribution of exogenous ROS compared to endogenous mitochondrial ROS can only be estimated indirectly. One of the evaluation methods can be a comparison of basal and induced proton leakage. Basal leakage is relatively constant at a given level of metabolism, whereas the latter correlates with UCP activity under stress. The corresponding comment has been added to section 4.1.1 on p.7.

Point 3: On p9 l369-372 the authors mention an interaction between Nrf2 and Mitf but only say “Interactions between these transcription factors control melanogenesis, mitochondrial biogenesis, and redox homeostasis [135].” It would greatly help the reader to be told what this interaction consists of and how it actually controls the listed functions.  More depth is always helpful in an article of this type.

Response 3: Add some information about interaction between Nrf2 and Mitf (section 4.2.1, p.9).

Point 4: Though mentioned briefly, and a number of references discuss it, the whole topic of RPE and aging is not dealt with adequately.  A separate section summarizing what we know about RPE and aging, and its relationship with diseases such as ARMD, would be very valuable.

Response 4: We added a separate section about RPE aging with relationship with ARMD (AMD) and oxidative stress (section 6.1, p. 17-18).

Reviewer 2 Report

The authors have compiled a comprehensive overview of the role of oxidative stress and antioxidants as therapy in RPE-based pathologies. The content itself seems good, with relevant studies cited. The language used is oftentimes difficult to understand or incorrect; moderate editing is required.

Just some minor points:

- It should be BM, and not MB.

- L103- I would not call it single-row but monolayer

- Not all abbreviations named in Figure legends. Text very small to read.

- L499 - et al used instead of etc

As mentioned above, some editing is required

Author Response

Response to Reviewer 2 Comments

General comment: The authors have compiled a comprehensive overview of the role of oxidative stress and antioxidants as therapy in RPE-based pathologies. The content itself seems good, with relevant studies cited. The language used is oftentimes difficult to understand or incorrect; moderate editing is required.

Response to general comment: Now we are correcting the language of our article, but, unfortunately, we will need additional time for this. The corrected version of the text will be sent to you in the nearest time.

Point 1: It should be BM, and not MB.

 Response 1: We corrected MB to BM (Bruch’s membrane) through all text.

Point 2:. L103- I would not call it single-row but monolayer

Response 2: We agreed and changed single-row to monolayer

Point 3: Not all abbreviations named in Figure legends. Text very small to read.

Response 3: We named all abbreviations in Figure legends. The size of fonts of figures was been enlarge to improve reading.

Point 4: L499 - et al used instead of etc

Response 4: Changed et al. to etc.

Additional response to Reviewer 2 Comments: We have adjusted the language of our article using the MDPI language editing service. Now we are sending you this text with corrections and comments.

Reviewer 3 Report

The paper of Yulina and Vladimir analyzes the mechanisms underlying redox control in the retinal pigment epithelium and its importance in the prevention of ocular diseases. The review addresses the various aspects in a thorough and clear way by analyzing the molecular mechanisms in detail. The only observation concerns the possibility of discussing in chapter 4 the pro-oxidant systems first and then the anti-oxidant systems, including autophagy among these. Overall I think it can be accepted for publication with minor revision.

Author Response

Response to Reviewer 3 Comments

General comment: The paper of Yulina and Vladimir analyzes the mechanisms underlying redox control in the retinal pigment epithelium and its importance in the prevention of ocular diseases. The review addresses the various aspects in a thorough and clear way by analyzing the molecular mechanisms in detail. Overall I think it can be accepted for publication with minor revision.

Point 1: The only observation concerns the possibility of discussing in chapter 4 the pro-oxidant systems first and then the anti-oxidant systems, including autophagy among these.

 Response 1: We agree that autophagy have to discuss with key components of the antioxidant defense system in RPE cells. Therefore, we moved the autophagy to the section “Key components of the antioxidant defense system in RPE cells” (section 4.2.8. Autophagy, p.12).